# Northern bobwhite select for shrubby thickets interspersed in grasslands during fall and winter

Alisha R. Mosloff[1]*, Mitch D. Weegman[1], Frank R. Thompson[2], Thomas R. Thompson[3]

1 School of Natural Resources, University of Missouri, Columbia, Missouri, United States of America,
2 USDA Forest Service Northern Research Station, Columbia, Missouri, United States of America,
3 Missouri Department of Conservation, Clinton, Missouri, United States of America

☯ These authors contributed equally to this work.
* arm94b@mail.missouri.edu

**Data Availability Statement:** The data underlying the results of this study are available in the United States Forest Service data repository at the following link: https://doi.org/10.2737/RDS-2021-0040.

## Abstract

Resource selection is a key component in understanding the ecological processes underlying population dynamics, particularly for species such as northern bobwhite (*Colinus virginianus*), which are declining across their range in North America. There is a growing body of literature quantifying breeding season resource selection in bobwhite; however, winter information is particularly sparse despite it being a season of substantial mortality. Information regarding winter resource selection is necessary to quantify the extent to which resource requirements are driving population change. We modeled bobwhite fall and winter resource selection as a function of vegetation structure, composition, and management from traditionally (intensively) managed sites and remnant (extensively managed) grassland sites in southwest Missouri using multinomial logit discrete choice models in a Bayesian framework. We captured 158 bobwhite from 67 unique coveys and attached transmitters to 119 individuals. We created 671 choice sets comprised of 1 used location and 3 available locations. Bobwhite selected for locations which were closer to trees during the winter; the relative probability of selection decreased from 0.45 (85% Credible Interval [CRI]: 0.17–0.74) to 0.00 (85% CRI: 0.00–0.002) as distance to trees ranged from 0–313 m. The relative probability of selection increased from near 0 (85% CRI: 0.00–0.01) to 0.33 (85% CRI: 0.09–0.56) and from near 0 (85% CRI: 0.00–0.00) to 0.51 (85% CRI: 0.36–0.71) as visual obstruction increased from 0 to 100% during fall and winter, respectively. Bobwhite also selected locations with more woody stems; the relative probability of selection increased from near 0.00 (85% CRI: 0.00–0.002) to 0.30 (85% CRI: 0.17–0.46) and near 0.00 (85% CRI: 0.00–0.001) to 0.35 (85% CRI: 0.22–0.55) as stem count ranged from 0 to 1000 stems in fall and winter, respectively. The relative probability of selection also decreased from 0.35 (85% CRI: 0.20–0.54) to nearly 0 (85% CRI: 0.00–0.001) as percent grass varied from 0 to 100% in fall. We suggest that dense shrub cover in close proximity to native grasslands is an important component of fall and winter cover given bobwhite selection of shrub cover and previously reported survival benefits in fall and winter.

**Funding:** This work was funded by a cooperative agreement between the Missouri Department of Conservation and University of Missouri-Columbia, Project 00055731, F. Thompson III and M. Weegman principal investigators. The University of Missouri Columbia administered funding and provided logistical and administrative support. Further in-kind and logistical support was provided by the USDA Forest Service Northern Research Station. The funders provided comments on the study design, data collection and analysis, and a draft of the manuscript but decisions on revisions and to publish were made by the authors.

**Competing interests:** The authors have declared that no competing interests exist.

## Introduction

Resource selection is a key component in understanding the ecological processes underlying population dynamics [1]. Conservationists may gain better understanding of factors driving abundance at various spatial and temporal scales by linking the impacts of predation, foraging behavior, and availability of resources [1]. The resource requirements of species such as upland game birds may change over the course of their life cycle (e.g., from juveniles to adults) and annual cycle (e.g., from the breeding season to the non-breeding season). Thus, conservation practitioners require an understanding of resource selection throughout the annual cycle to fully understand the trade-offs in life history events and land use. There is a growing body of literature on breeding season resource selection of upland game birds [2–5], especially for northern bobwhite (*Colinus virginianus*, hereafter bobwhite) [6–10]. However, many bobwhite populations are declining [11], suggesting that current understanding and conservation plans based on resource selection during the breeding season is not enough to effectively increase bobwhite populations [12]. Information regarding fall and winter resource selection is necessary for more complete understanding of habitat requirements in each season to refine conservation decisions throughout the annual cycle.

Grasslands and the associated bare ground, forbs, and shrubs provide foraging and loafing cover for bobwhites [13,14]. Grasslands dominated by native warm season grasses are particularly beneficial for bobwhite as they provide greater quality habitat than non-native grasses if they are managed by burning, grazing, herbicide, or disking to keep them from becoming overly tall and dense which renders them unsuitable for bobwhite [15–18]. Declines in bobwhite abundance are typically attributed to losses of grasslands that are converted to agricultural crops [19,20], suggesting that landscapes with greater quantities of grasslands could increase bobwhite abundance. However, previous studies have found decreased selection for grasslands and agricultural crops as the availability of these vegetation types increases [21,22]. This likely indicates a functional response of selection relative to the abundance of either vegetation type, suggesting that the benefits of either vegetation type diminish as they become increasingly available [22,23].

Bobwhite are a shrub-obligate species that require dense, early successional, woody vegetation interspersed on the landscape in a manner that it is immediately accessible for use as escape cover from predators, thermal refugia, protection from extreme winter weather, and safe loafing areas [21,24–26]. However, unmanaged shrubs can quickly succeed to forest. While forests can provide escape cover and loafing sites when scrub-shrub is limited [27], trees provide habitat for predators of bobwhite [28] and closed canopy forests (i.e., canopy cover $\geq$60%) often limit production of grasses, forbs, and shrubs [29,30]. Janke and Gates found that early successional woody vegetation was selected for over other vegetation types, while forests were generally not selected for and that the value of other vegetation types, such as grasslands, are maximized as smaller patches in close proximity to woody vegetation [22]. Heterogeneity results from the interspersion of small patches of different vegetation types. Of particular relevance to bobwhite is the heterogeneity created by interspersion of grassland or agriculture and woody vegetation; as this heterogeneity increases, so does proximity to woody vegetation. Generally, bobwhite select for habitat patches which average at least 30% early successional woody cover [31–33]. The National Bobwhite Conservation Initiative (NBCI) Coordinated Implementation Plan (CIP) considers locations $\geq$50 m from woody vegetation as unsuitable habitat for bobwhite [34].

Bobwhite management typically focuses on creating heterogeneous landscapes which maximize usable space. However, the structural attributes of the vegetation types (e.g., height, density, and visual obstruction) within these landscapes are also important [30,33,35–37]. Winter

feeding areas are most beneficial when vegetation is interspersed with 25–60% bare ground [38]. Bobwhite roost sites in Missouri typically consist of 27% forb, 23% bare ground [39], 65% litter, and a maximum vegetation height of 94 cm [40]. Sites which offer high visual obstruction provide enhanced protection from predators [33]. Kopp et al. found that bobwhite used habitat with bare ground ranging from 10 to 60% and herbaceous vegetation ranging from 0 to 35% [37]. The utility of these structural attributes is maximized when they are interspersed across the landscape in patches of single or multiple attributes, increasing the diversity of the habitat.

Heterogeneous landscapes are often created through disturbance, such as prescribed fire, grazing, and disking, and are often distributed non-uniformly in space and time [41]. Resource selection of bobwhite is impacted by these disturbances. Previous studies have documented differences in resource selection of bobwhite on managed, publicly-owned lands and privately-owned lands dominated by agricultural production [42,43]. Activities such as the harvest of row crops may lead to the creation of unusable space in the winter months [44,45]. Prescribed fire applied during winter may create unusable space for several months post-burn through the removal of grass, but creates bare ground for improved mobility for a couple of years post-burn [46,47] and increases forage (i.e., seeds) abundance and availability [48]. Similarly, grazing promotes the complex vegetation structure required by bobwhite as well as the amount of litter and bare ground consistent with the habitat requirements of bobwhite [49]. Historically, disturbance throughout the bobwhite range consisted of periodic, low-intensity fire [50] and ungulate grazing [51]. The extent of the heterogeneity of a landscape may be quantified through multiple methods. We quantified heterogeneity through the use of percent cover and distance to a given vegetation type because intermediate levels of percent cover of multiple vegetation types and small distances among vegetation types is indicative of heterogeneity.

Effective conservation strategies require detailed knowledge of resource selection patterns of bobwhite. Managers typically adopt intensive, or traditional, management regimes for bobwhite. Intensive management regimes mimic the formerly fragmented landscapes of the US through the use of food plots, native grasses, and shrub rows closely interspersed throughout the landscape. Food plots of intentionally planted row crops such as corn, wheat, and soybeans provide forage for breeding adults and chicks during the summer, managed native grass provides bare ground and forage throughout the year, and shrub/tree rows provide woody escape cover during winter. However, extensive management regimes have been developed in large native grassland landscapes and utilize combinations of prescribed burning, grazing, mechanical brush removal, haying, high clipping, and herbicide. Extensive management tries to mimic pre-settlement landscapes that meet the yearly resource requirements of bobwhite. Prescribed burning and grazing mimic wildfires and bison grazing of pre-settlement times and remove vegetation and litter from the area creating bare ground. Mechanical brush removal prevents succession of shrubs to trees and creates litter from cut woody vegetation. Haying and high clipping reduce the height of grasses and forbs; however, haying removes litter while high clipping does not. Herbicides are used selectively to control undesirable vegetation. The spatial configuration of these factors likely impacts resource selection of bobwhite [52,53].

A modeling framework that allows robust evaluations of seasonal resource selection would help improve our understanding of seasonal resource needs. Discrete choice models allow comparison of used and available vegetation types even when availability changes over time (e.g., due to harvest of row crops or prescribed burning) [54,55]. We used discrete choice models to directly evaluate the impacts of vegetation type, vegetation structure, and management on bobwhite resource selection.

Our objectives were to quantify resource selection of bobwhite during fall and winter for vegetation characteristics and management treatments. Many ecological factors, operating at different scales, may influence the resource selection of bobwhite. Hernandez and Guthery found that herbaceous and woody cover should be interspersed in such a way that an individual is never further then 30 m from either vegetation type [56]. Further, the National Bobwhite Conservation Initiative (NBCI) Coordinated Implementation Plan (CIP) does not consider vegetation types further than 50 m from woody vegetation as suitable habitat [57]. Thus, we evaluated both the structure and composition of vegetation for selection preferences in bobwhite at a local scale (50 m). We fit multinomial logit discrete choice models representing competing hypotheses within a model selection framework to compare support for hypotheses relating to native grass, managed grasslands, woody vegetation, and trees. We hypothesized that bobwhite coveys would select for locations which consisted of 1) greater proportions of native warm season grass, 2) greater proportions of grasslands which were managed via burning, grazing, or both within the previous 12 months, 3) landscapes which contained greater woody edge density, 4) locations further from trees, and 5) locations which provided structure with moderate amounts (25–60%) of forbs, grasses, and bare ground, and high visual obstruction (>20 cm).

## Materials and methods

### Study area

We conducted our research on five Missouri Department of Conservation (MDC) conservation areas in southwest Missouri: Robert E. Talbot Conservation Area (Talbot; 37.1893257, -93.9295361), Shawnee Trail Conservation Area (Shawnee Trail; 37.4352157, -94.5622991), Stony Point Prairie Conservation Area (Stony; 37.5364583, -94.0037019), Wade and June Shelton Memorial Conservation Area (Shelton; 37.462208, -93.984749), and Wah'Kon-Tah Prairie Conservation Area (Wah'Kon-Tah; 37.8977311, -93.9941023; Fig 1). This region has a high potential for successful bobwhite habitat management and population recovery in Missouri [57]. Native prairie sites were dominated by big bluestem (*Andropogon gerardii*), indiangrass (*Sorghastrum nutans*), and little bluestem (*Schnizachyrium scoparium*). More traditional conservation areas included warm season grass and native prairie plantings, in addition to cool season and mixed grass pastures and fields that contained large quantities of tall fescue (*Festuca acundinacea*). Dominant shrub and trees included sumac (*Rhus* spp.), blackberry (*Rubus* spp.), plum (*Prunus* spp.), dogwood (*Cornus* spp.), and oak (*Quercus* spp.). The surrounding

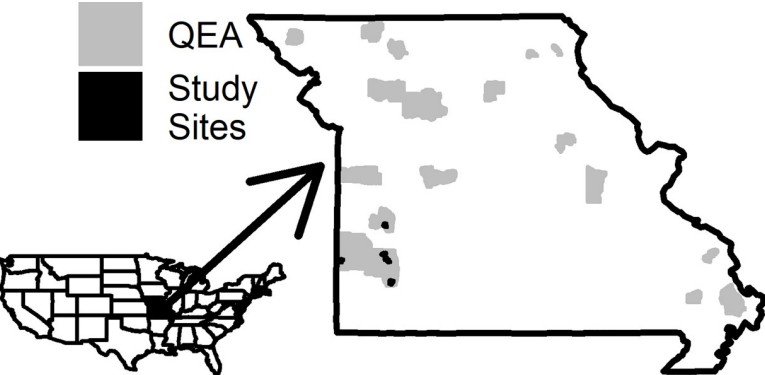

**Fig 1.** Quail emphasis areas (grey) and study sites (black) where we examined northern bobwhite fall (Nov) and winter (Dec–Jan) resource selection in southwest Missouri, 2017–2019.

counties (Barton, Cedar, Dade, Lawrence, and St. Clair) were largely converted to non-native pasture (tall fescue), deciduous forest, cultivated crops (e.g., corn, soybeans, wheat), and urban development [58,59].

Sites varied in history, size, and management strategy. Shelton (130 ha), Stony (390 ha), and Wah'Kon-Tah (944 ha) were predominantly remnant native tallgrass prairies extensively managed with a combination of fire, grazing, mechanical brush removal, haying, high clipping (mowing grasses to a height ≥30 cm), and herbicide to maintain a grassland landscape which mimics pre-settlement landscapes. Grazing, mowing, and prescribed burns were used to mimic the pyric herbivory characteristic of historic disturbance and enhance the species richness and vegetation structure of the area [60,61]. Talbot and Shawnee Trail were intensively (traditionally) managed sites (1764 and 1471 ha, respectively) composed of small-scale row-crop (i.e., corn, soybeans, wheat) agriculture, food plots (i.e., corn, soybeans, wheat intentionally planted and left standing for wildlife), woody vegetation, and planted grassland managed by prescribed grazing, mowing, and prescribed burning which mimic post-settlement, fragmented landscapes. Agricultural practices occurred historically on Shawnee Trail and cattle grazing occurred on Talbot prior to land acquisition by the MDC. Several of the agricultural units have since been restored to native grass and mixed-prairie on both Shawnee Trail and Talbot.

All sites were managed with prescribed burning and grazing (in combination or separately) throughout the duration of the study. Average burn unit sizes were generally larger on extensively managed sites (approximately 10–40 ha) than intensively managed sites (approximately 4–16 ha). Prescribed burning typically occurred on a 3- to 5-year rotation with extensively managed sites burned more frequently than intensively managed sites. Prescribed burning occurred from January to April and August to December. Prescribed burn units which were managed concurrently with prescribed grazing (i.e., patch burn grazing) were generally burned February to April. Managed sites with remnant or restored prairie are typically grazed at a stocking rate of 1 animal unit (AU) per 2.2 ha where 1 AU is 1,000 pounds of cattle with grazing durations ranging from 90–120 days. Managed sites that were not prairie, but primarily comprised of warm season grass plantings and non-native cool season grass pastures, were grazed at a higher intensity of 1 AU per 1.6 ha with grazing durations ranging from 90 to 120 days (K. Hedges, Missouri Department of Conservation, personal communication).

## Captures and tracking devices

We captured bobwhite using funnel traps baited with cracked corn and milo in October 2018 [13]. We located capture sites where bobwhite were observed by field staff and near locations of bobwhite radio tagged in an ongoing breeding season study. We attempted to mark individuals from as many unique coveys as possible. Capture sites were pre-baited for 4 to 10 days prior to placement of a funnel trap. Funnel traps were weighed down with bricks and covered with eastern red cedar (*Juniperus virginiana*) boughs to prevent predators from locating trapped bobwhite and to provide protection from inclement weather. We checked traps 2 to 4 hours after sunrise and no earlier than 30 min before sunset to reduce the amount of time birds spent in traps. We did not trap if severe weather (e.g., heavy rainfall) was expected that could harm trapped individuals. We removed a trap from a site if any individual was captured 3 times to reduce exposure to trapping. We ceased trapping and removed all evidence of trapping (i.e., bait, cedar boughs, and flagging) prior to November 1 each year because that was the opening of the bobwhite hunting season in Missouri. Up to four individuals per covey were fitted with transmitters to spread units among coveys.

We marked bobwhite with a uniquely numbered leg band (National Band & Tag Company, Newport, KY, US) and individuals >100 g were fitted with a pendant-style radio transmitter

that weighed 5.3–5.5 g (model AWE-QII from American Wildlife Enterprises, Monticello, FL, US). We recorded sex, age class (adult/juvenile) [62,63], weight (g), and tarsus length (mm) and released individuals at their capture site. While attachment of transmitters may handicap birds using marginal habitats [25], studies based on large samples of marked individuals with rigorous mark-recapture methods have not detected a negative effect of tracking devices on bobwhite survival [64–66]. All animal procedures were approved under University of Missouri Animal Care and Use Committee protocol #8766.

## Quail locations

We attempted to locate bobwhite three times per week from 1 November 2018 to 31 January 2019 by homing to within 10 m of individual [67]. We recorded our location with a Global Positioning System (GPS) unit, as well as the distance (m) and azimuth to the bobwhite. One location was recorded for coveys and individuals were assumed to be in the same covey if they were within 10 m of each other. We used triangulation from two or more points [67] to locate individuals on property we did not have access to. We attempted to not flush individuals.

After 12 hours of no movement the transmitters emitted a fast pulse rate at which point we would locate the transmitter and individual, and classified it as either a dropped transmitter or mortality. If a transmitter was located with no apparent damage and no sign of predation, it was considered a dropped collar and included in the analysis as alive to that day. Mortality events were ascribed when a collar showed evidence of damage, was located with a dead bobwhite, or near sign of a predator (e.g., predator was observed with carcass, transmitter was in an animal burrow, or located in scat). Individuals that were classified as a drop or mortality within the first seven days of the tracking period were eliminated from the dataset. We recorded the vegetation classification, management that had been conducted, and if the individuals were in a tree edge, shrub, or both at each location. We searched for individuals that we could not locate by radio telemetry at least once per week by intensively searching the area surrounding the last known location and by searching the larger, surrounding area using a dipole omnidirectional antenna on local roads.

## Vegetation and management classifications

Maps of vegetation and management types were provided by land managers for each study site. We considered up to 13 unique vegetation types: agriculture (planted row crops typically corn, wheat, soybeans), cool season grass, food plot (small patches of row crops planted and left un-harvested for wildlife), idle agriculture (agricultural fields not planted in the prior growing season), mixed grass (both cool and warm season grasses), native grass planting (native grasses were planted into existing vegetation), native prairie (undisturbed, virgin prairie), restored prairie (areas where non-native vegetation was restored to native prairie), oak savanna, shrub-scrub, strip crop (linear strips of agriculture), deciduous forest, and woodland. Management was identified as: bullhogged (mowed shrubs and litter left on site), grazed, hayed (grass and forbs are mowed and litter is removed), mowed (grasses and forbs are mowed and litter is not removed), prescribed burn, sprayed (treated with herbicide), and mechanical woody removal (similar to bullhogging, shrubs and trees are mowed or cut and litter is left on site). We improved mapping of shrubs and trees by using airborne light detection and ranging (LiDAR) data from years 2014 (Vernon and Barton county) and 2016 (Bates, St. Clair, Cedar, Dade and Lawrence counties) that were publicly available from the Missouri Spatial Data Information Service (http://www.msdis.missouri.edu). Shrubs, trees/tree edges, and woodlands are often associated with ground cover consisting of grasses and forbs. We created separate rasters at 3.6 m resolution to represent shrubs and trees by classifying vegetation

height from 0.7 to 3.5 m as shrubs and 3.5 to 40 m as trees [68]. Ecological factors operating at different scales influence bobwhite habitat occupancy and survival [69]. Therefore, we assessed vegetation metrics at the covey's location and vegetation and management metrics both at the covey's location and within a 50 m radius of the location. We condensed vegetation and management types in ArcGIS [70] to ecologically meaningful categories that directly addressed our hypotheses concerning native grass, prescribed burning, and prescribed grazing. Given that native warm season grasses provide higher quality habitat for bobwhite than non-native grasses such as fescue (*Festuca arundinacea*) [16,17] we hypothesized that bobwhite would select for locations with greater amounts of native grass. We categorized native grass as any parcels within a conservation area that were native prairie, native grass planting, and restored prairie identified as primarily native grass via field verification, regardless of associated management technique. To evaluate our hypothesis that bobwhite would select locations in native grass, we calculated the proportion of land within 50 m of a covey's location that was native grass (NG). However, unmanaged native grass quickly loses value for bobwhite, so we also modeled the proportion of land that was managed via prescribed burning and grazing (i.e., combined all land that had been prescribed burned and/or grazed at any time during the study). To evaluate our hypothesis that bobwhite would select locations managed via prescribed burning and grazing, we calculated the proportion of land managed with either prescribed burning or grazing, or both, within the last 12 months and within 50 m of a covey's location (PG). We believe the amount of woody vegetation (i.e., shrubs and trees) on the landscape as well as the interspersion of this woody vegetation throughout the landscape may impact fall and winter resource selection. To evaluate our hypothesis that bobwhite select for locations with higher interspersion of woody cover, we created a variable which quantified the amount of woody edge (i.e., shrubs and trees) within 50 m of the covey's location (WE) whereby a higher woody edge density would represent higher interspersion of woody vegetation. We also calculated the distance from the nearest tree (m; TD) to evaluate our hypothesis that bobwhite would select locations further from trees. We calculated all landscape metrics (e.g., percent cover, distance from tree) using the landscapemetrics package [71] in program R version 3.6.0 [72].

## Vegetation measurements

We measured vegetation weekly at one location per covey stratified by time of location to ensure vegetation data were representative of sites used during the entire day. To account for daily temporal variation in resource selection, one third of the vegetation measurements each week were collected at roost locations (16:30–08:30), one third were collected at morning locations (08:30–12:30) and one third were collected at afternoon locations (12:30–16:30). Individuals located within 10 m of each other were considered to be in the same covey and the location was determined to be where the majority of marked individuals within the covey were located. Marked individuals further than 10 m apart were considered unique coveys and given separate locations at which a unique set of vegetation measurements were collected. Vegetation measurements also were collected at three available locations because our resource selection modeling approach required data for each used location be paired with available locations which were available for selection at the time the covey chose the used location. We randomly chose three available locations by distance and azimuth from the used location within 400 m of the used location. Random locations were generated using a random number generator in Microsoft Excel [73] whereby distances were constrained between 50 and 400 m to ensure random locations did not overlap and azimuths were constrained between one and 360. We chose 400 m as the maximum distance because it was the 90% quantile of distance traveled by

bobwhite within our study between resightings and we assumed points within this radius were available to bobwhite.

Vegetation composition was quantified using a 0.50-m$^2$ quadrant [modified from 74] where we estimated percent of forbs (FB), grasses (GS), agricultural crops (AG; e.g., corn, wheat, soybeans), woody vegetation (W) and bare soil (BR), in 4% cover classes within the Daubenmire frame. We estimated visual obstruction with a 2-m modified robel pole [75] divided into 20, 10-cm segments read from 5 m away and 1 meter above the ground in each of the four cardinal directions, and averaged the readings to quantify visual obstruction (VO). We counted the number of woody stems (SC), including trees, >1 m in height and within 5 m of each location.

## Discrete choice analyses

We fit conditional multinomial logit discrete choice models in a Bayesian framework [76] implemented in JAGS [77] using Program R to model the probability that a covey would select a location given a choice between three locations available at one time (i.e., forming a choice set) [54,78]. Discrete choice models allow for comparison of covariates within choice sets rather than comparisons of used and available locations across all locations and times. Further, the composition of a choice set is allowed to change over time and between coveys, removing the variation in factors which often influence selection (e.g., an individual's age or the availability of habitat) [55]. Individual bobwhite form social groups, or coveys, during the fall and winter. We believe that an individual bobwhite's resource selection decisions may be influenced by the other individuals in the covey. A covey identification number (covey ID) was assigned to each unique combination of individuals located within 10 m of each other at the time of resighting. We assigned a single, common location representing the location of the majority of individuals within the covey to all individuals in a covey. Individuals located further than 10 m from the covey were considered a unique covey and assigned a unique location. We standardized all covariates prior to model runs and assessed multicollinearity among variables by calculating variance inflation factors (VIF) using the car package [79] in Program R version 3.6.0 [72]. We only considered combinations of variables in a model that resulted in VIF values <2.5 [80,81].

We considered the probability that a covey, $c$, would select the used or available locations if given a choice between four locations (i.e., one used, three available). We modeled the 'utility' of each used location in the $i$th choice set of bobwhite covey $c$ as a linear function of covariates representing the vegetation structure and the composition of surrounding vegetation and management types, following Jenkins et al. [76]:

$$U_{ic}^{used} = \beta_{1c}FB_{ic}^{used} + \beta_{2c}GS_{ic}^{used} + \beta_{3c}BR_{ic}^{used} + \beta_{4c}VO_{ic}^{used} + \beta_{5c}SC_{ic}^{used} + \\ \beta_{6c}NG_{ic}^{used} + \beta_{7c}PG_{ic}^{used} + \beta_{8c}NG_{ic}^{used}PG_{ic}^{used} + \beta_{9c}TD_{ic}^{used} + \beta_{10c}WE_{ic}^{used} \tag{1}$$

where $FB_{ic}^{used}$ is the percentage of forbs within a 0.50-m$^2$ quadrant frame at the used location, $GS_{ic}^{used}$ is the percentage of grasses within a 0.50-m$^2$ quadrant frame at the used location, $BR_{ic}^{used}$ is the percentage of bare ground within a 0.50-m$^2$ quadrant frame at the used location, $VO_{ic}^{used}$ is the averaged visual obstruction from the 4 cardinal directions (cm), $SC_{ic}^{used}$ is the amount of woody stems >1 m tall within 5 m of the used location, $NG_{ic}^{used}$ is the percent of native grass vegetation type within 50 m of the used location, $PG_{ic}^{used}$ is the percent of land managed using either or both prescribed fire and grazing within 50 m of the used location, $NG_{ic}^{used}PG_{ic}^{used}$ the interactive effect of $NG_{ic}^{used}$ and $PG_{ic}^{used}$, $TD_{ic}^{used}$ is the distance from the nearest tree edge from the used location, and $WE_{ic}^{used}$ is the density of shrub and tree edges within 50 m of the used

location, and $\beta_{1c}, \ldots, \beta_{10c}$ are the covey-level coefficients corresponding to covey $c$ ($c \, \epsilon \, (1, 2, \ldots, C)$, where $C$ is the total number of coveys included in the model. Combinations of covariates included in individual models were selected based on which covariate combinations were identified as important to bobwhite survival in an ongoing study [82]. We modeled the utility of each available location in an identical manner, substituting covariates at used locations for covariates at available locations. We then used the utility functions defined above to model the probability of selecting the used or available locations when given a choice among the four locations (hereafter relative probability). We calculated the relative probability of selecting a location as:

$$P_n(j) = \frac{\exp(U')}{\sum_{k \epsilon j_n} \exp(U')} \tag{2}$$

where $P_n(j)$ is the probability of location $j$ being chosen by a given covey for it's $n$th selection event, assuming that each selection event is independent of previous selection events [54].

We modeled population level resource selection of bobwhite by assuming that covey-level coefficients arose from Normal population-level distributions [78,83]:

$$(\beta_k) \sim N(\mu_k, \sigma_k^2), \tag{3}$$

where $\mu_k$ is the population mean and $\sigma_k^2$ is the variance for the effect of covariate $k$ ($k \, \epsilon \, (1, 2, \ldots, K)$ where $k$ is the covariate of interest representing our hypothesized drivers of resource selection. We hereafter refer to each regression coefficient distribution by the name of the associated vegetation and management covariates.

We hypothesized that resource selection would vary across months due to changes in weather and vegetation structure (i.e., vegetation senescence and decomposition as winter progresses) and initially considered how resource selection varied as a function of each month during fall-winter (i.e., November, December, January). However, models would not converge due to sparse data so we then defined a binary variable to represent fall or winter (i.e., November = 0, December and January = 1) and modeled separate population-level parameter distributions for fall and winter. Similarly, a global model including all covariates would not converge due to sparse data so we do not report the results from that model. Preliminary analyses indicated models including season were more supported so we included season as a fixed effect in all subsequent models.

We used Markov chain Monte Carlo (MCMC) algorithms to estimate the posterior distributions of each parameter [77]. We ran 4 chains for 100,000 iterations after a 10,000 iteration burn-in and a thinning of 50 with vague priors and assumed normal prior distributions ($N \sim (0, 0.01)$) on all vegetation and management regression coefficients and normal prior distributions ($N \sim (0, 0.01)$) for each population-level mean hyperparameter (i.e., a parameter of the prior distribution which allowed for inclusion of a random effect for covey). We evaluated model convergence by visual inspection of MCMC chains and ensured the Gelman-Rubin convergence statistic was <1.1 [84].

We compared support among models based on the Watanabe-Akaike Information Criteria (WAIC) [84,85] and interpreted results from models with $\Delta$WAIC < 2. We removed more complex competitive models (within <2 WAIC units) from further consideration in favor of simpler models that shared one or more of the same terms to eliminate uninformative models [86]. For each covariate coefficient we present the mean of the posterior distribution (PM) and 85% credible interval (CRI). In the Bayesian framework, the proportion of the posterior distribution that is positive or negative represents the probability of a positive or negative effect, respectively. In addition to the mean effect and 85% CRI, we also report the proportion of

posteriors that were positive or negative when >0.85 of the distribution was above or below zero, as another measure of support. We estimated and interpreted relative probability of use curves over the observed range of covariates of interest, while holding other covariates at their means [55]. Relative probability of use predictions are presented based on the mean, 7.5%, and 92.5% posterior values of the covariate of interest to show variability in the mean effect size.

## Results

We captured 158 bobwhite from 67 unique coveys and attached transmitters to 119 individuals in 2018. We used locations from 9, 23, 6, and 19 adults on Shawnee Trail, Stoney, Talbot, and Wah'Kon-Tah, respectively, and 6, 3, 8, 10, and 8 juveniles on Shawnee Trail, Shelton, Stoney, Talbot and Wah'Kon-Tah, respectively. These individuals comprised 10 unique coveys on Shawnee Trail, 2 unique coveys on Shelton, 11 unique coveys on Stoney, 10 unique coveys on Talbot, and 14 unique coveys on Wah'Kon-Tah. We obtained 2,595 locations during 2018. We collected vegetation data on 650 used locations and their associated three random locations for a total of 650 choice sets in our discrete choice analyses.

Comparisons of descriptive statistics suggested the mean percent grass at used locations was less than at available locations while the mean percent forb and percent bare ground were similar between used and available locations (Table 1). Mean visual obstruction was greater at used locations than available locations, as was mean stem count. Mean percent native grass and percent of land managed with prescribed burning and/or grazing in the last 12 months within 50 m were similar between used and available locations. Mean distance to tree at used locations was less than at available locations while mean woody edge density was greater at used sites than available locations (Table 1).

We ranked our six candidate models based on WAIC and the most supported model was model 1 (m1) while the null model had the least support (Table 2). We based inferences on the top model (m1). The second ranked model (m2) included unique and significant effects (i.e., percent forb and percent bare ground in winter) which were not interpreted due to ΔWAIC < 2 (Table 2). In general, we found support for our hypotheses that bobwhite selected greater densities of woody edges, higher woody stem counts, closer to trees, and greater

**Table 1. Minimum, mean, and maximum for all covariates used in discrete choice resource selection models for Northern bobwhite during fall (Nov) and winter (Dec–Jan; fall/winter) in southwest Missouri, 2018–2019.**

| Variable[a] | Used | | | Available | | |
|---|---|---|---|---|---|---|
| | Mean | Minimum | Maximum | Mean | Minimum | Maximum |
| GS | 25.82/22.36 | 0.00/0.00 | 100.00/100.00 | 39.68/29.38 | 0.00/0.00 | 100.00/100.00 |
| FB | 12.75/5.49 | 0.00/0.00 | 90.00/80.00 | 10.09/5.68 | 0.00/0.00 | 100.00/100.00 |
| BR | 7.95/6.40 | 0.00/0.00 | 80.00/95.00 | 9.11/10.73 | 0.000.00 | 100.00/100.00 |
| VO | 52.64/43.53 | 1.00/2.75 | 100.00/100.00 | 37.77/21.75 | 0.00/0.00 | 100.00/92.50 |
| SC | 74.91/69.03 | 0.00/0.00 | 800.00/500.00 | 25.50/15.38 | 0.00/0.00 | 500.00/1000.00 |
| NG | 0.48/0.40 | 0.00/0.00 | 1.00/1.00 | 0.53/0.44 | 0.00/0.00 | 1.00/1.00 |
| PG | 0.50/0.39 | 0.00/0.00 | 1.00/1.00 | 0.51/0.42 | 0.000.00 | 1.00/1.00 |
| TD | 47.56/45.48 | 0.00/0.00 | 194.03/295.10 | 65.62/69.55 | 0.00/0.00 | 312.99/304.31 |
| WE | 425.40/483.50 | 0.00/0.00 | 1563.6/1692.20 | 356.88/340.92 | 0.00/0.00 | 1628.04/2099.27 |

[a]Percent grass (GS), percent forb (FB), and bare ground (BR) within the 0.50 m² quadrat at the given location; visual obstruction measured by a modified Robel pole (VO); number of woody stems greater than 1 m in height and within 5 m of the given location (SC); percent of native grass within 50 m of the given location (NG); percent land managed with prescribed burning and/or grazing within 50 m of the given location (PG); distance (m) from nearest tree edge or individual tree (TD); density (m/ha) of shrub and tree edges within 50 m of the given location (WE).

**Table 2. Variables included in candidate models explaining fall (Nov) and winter (Dec–Jan) resource selection of Northern bobwhite in southwest Missouri, 2018–2019.** Variables with the proportion of the posterior distribution <0.85 are indicated with a "0", and variables with a proportion of the posterior distribution >0.85 are indicated with "+" if the effect was positive or "-"if the effect was negative (fall/winter). Model support is indicated by the Watanabe-Akaike Information Criteria (WAIC).

| Model[a] | FB | GS | BR | VO | SC | NG | PG | NG*PG | TD | WE | WAIC | ΔWAIC |
|---|---|---|---|---|---|---|---|---|---|---|---|---|
| m1 | | -/0 | | +/+ | +/+ | 0/0 | | | 0/- | +/+ | 565.14 | 0.00 |
| m2 | 0/- | -/- | 0/- | +/+ | +/+ | | | | | | 834.79 | 269.65 |
| m3 | | | | +/+ | | | | | -/- | 0/+ | 1119.35 | 554.21 |
| m4 | | | | | | 0/0 | -/0 | 0/0 | -/- | 0/+ | 1377.78 | 812.64 |
| m5 | | | | | | -/- | 0/0 | 0/- | | | 1657.70 | 1092.56 |
| m6 | | | | | | | | | | | 1860.41 | 1295.27 |

[a]All models included covey ID as a random effect and season as a fixed effect with additional fixed effects are percent forb within the 0.50 m$^2$ quadrat at the given location (FB), percent grass within the 0.50 m$^2$ quadrat at the given location (GS), percent bare ground within the 0.50 m$^2$ quadrat at the given location (BR), visual obstruction measured by a modified Robel pole (VO), quantity of woody stems greater than 1 m in height and within 5 m of the given location (SC), percent of native grass within 50 m of the given location (NG), percent land managed with prescribed burning and grazing within 50 m of the given location (PG), distance (m) from nearest tree edge or individual tree (TD), density (m/ha) of shrub and tree edges within 50 m of the given location (WE).

vegetation visual obstruction than was available; however, there were seasonal (fall vs winter) differences in selection of these variables. We could not evaluate our hypotheses that bobwhite selected moderate amounts of grass, forb, and bare ground because models with quadratic terms did not converge. We instead evaluated linear effects of grass, forb, and bare ground. Bobwhite selected locations closer to trees in winter, contrary to our hypothesis that they would select locations further from trees. We found no support for our hypothesis that bobwhite would select locations based on the amount of native grass or grass grazed and/or burned within the last 12 months in the surrounding 50 m.

Based on our top ranked model (m1), bobwhite selected locations closer to trees in winter but not in fall (Table 3; Fig 2). The relative probability of selection decreased from 0.45 (85% CRI: 0.18, 0.75) to 0.00 (85% CRI: 0.00, 0.002) as the distance from trees increased from 0 to 313 m in winter (Fig 3). Bobwhite also selected locations with lower percentages of grass at the location during fall but not winter (Table 3; Fig 2). The relative probability of selection decreased from 0.35 (85% CRI: 0.20, 0.54) to nearly 0 (85% CRI: 0.00, 0.001) as grass varied from 0 to 100% in fall (Fig 3). Bobwhite selected locations with greater visual obstruction in winter, but not fall, and greater woody stems in fall and winter (Table 3; Fig 2). The relative probability of selection increased from near 0 (85% CRI: 0.00, 0.00001) to 0.29 (85% CRI: 0.36, 0.71) as visual obstruction increased from 0 to 100% in winter (Fig 3). The relative probability of selection increased from near 0.00 (85% CRI: 0.00, 0.003) to 0.30 (85% CRI: 0.17, 0.50) and near 0.00 (85% CRI: 0.00, 0.001) to 0.35 (85% CRI: 0.22, 0.55) as stem count ranged from 0 to 1000 stems in fall and winter, respectively (Fig 3). The credible interval for woody edge density overlapped zero; however, the proportion of the posterior distribution that was positive was 0.86 and 0.92 in fall and winter, respectively, indicating substantial support for a positive effect in each season (Table 3; Fig 2). The relative probability of selection increased from near 0.00 (85% CRI: 0.00, 0.11) to 0.27 (85% CRI: 0.00, 0.57) and from near 0.00 (85% CRI: 0.00, 0.10) to 0.19 (85% CRI: 0.00, 0.45) as woody edge density increased from 0 to 2100 m/ha in fall and winter, respectively.

## Discussion

Bobwhite resource selection was positively related to woody edge density (i.e., both tree and shrub edges), positively related to woody stem counts, indicative of shrubs, and negatively related to distance from trees; however, there were seasonal differences. The probability of

**Table 3. Posterior means and 85% credible intervals (CRI) for landscape, vegetation, and management effects from discrete choice models for fall (Nov) and winter (Dec–Jan) resource selection of Northern bobwhite in southwest Missouri, 2018–2019.**

| | | Model 1 | |
|---|---|---|---|
| Effect[a] | Season | PM | 85% CRI |
| GS | Fall | -3.77 | -6.28, -1.76 |
| | Winter | -0.25 | -1.23, 0.63 |
| VO | Fall | 2.48 | 0.47, 4.83 |
| | Winter | 4.59 | 2.63, 7.29 |
| SC | Fall | 2.67 | 1.29, 4.35 |
| | Winter | 3.35 | 1.72, 5.60 |
| NG | Fall | 0.21 | -2.73, 3.11 |
| | Winter | -0.29 | -2.61, 1.90 |
| TD | Fall | -0.92 | -3.17, 1.12 |
| | Winter | -3.28 | -6.54, -0.91 |
| WE | Fall | 1.37 | -0.44, 3.41 |
| | Winter | 0.87 | -0.02, 1.86 |

[a]Percent grass within the 0.50-$m^2$ quadrat at the given location (GS), visual obstruction measured by a modified Robel pole (VO), quantity of woody stems greater than 1 m in height and within 5 m of the covey's used location (SC), percent of native grass within 50 m of the given location (NG), distance (m) from nearest tree edge or individual tree (TD), density (m/ha) of shrub and tree edges within 50 m of the given location (WE).

selection was negatively related to percent grass available at the immediate location in fall while the proportions of native grass managed with prescribed burning and grazing did not influence the relative probability of selection. Our results support the findings of previous researchers that the structural attributes of vegetation, such as woody edge (shrub and tree) density and visual obstruction are important habitat features [30,33,35–37].

Bobwhite selected locations with greater quantities of woody stems, supporting our hypothesis that they would select locations with high densities of woody vegetation. High densities of woody stems can be indicative of the presence of woody vegetation such as shrub thickets. Brooke et al. found that the number of woody stems per hectare was not a strong predictor of non-breeding resource selection [9]; however, their analysis included woody stems >1.37 m in height whereas our analysis included woody stems >1 m in height. High woody stem densities may be created through re-sprouting of stems after mechanical brush removal, mowing, or prescribed fire, and maintained through the periodic disturbance at frequencies great enough to prevent woody vegetation from getting too tall or succeeding to trees, while not too frequent to maintain adequate interspersion of woody stems.

The amount and interspersion of woody vegetation is a major contributor to resource selection of bobwhite during the fall and winter [22]. Woody vegetation, especially shrubs, provide escape cover from predators and weather conditions [21,24–26]. We found weak support for the effect of woody edge density on resource selection. The use of areas characterized by higher woody edge density is consistent with findings in Illinois [87], Kansas [88], Kentucky [9], and Ohio [22]. Unger et al. hypothesized that bobwhite selected for locations near woody edges as the mixed vegetation at these locations maximized the resources available for forage and protective cover [89]. An increase in the amount and spread of woody vegetation across the landscape may lead to an increase in the probability of selection of that location by bobwhite and create more usable space. However, care should be taken to prevent succession of early successional woody vegetation into trees.

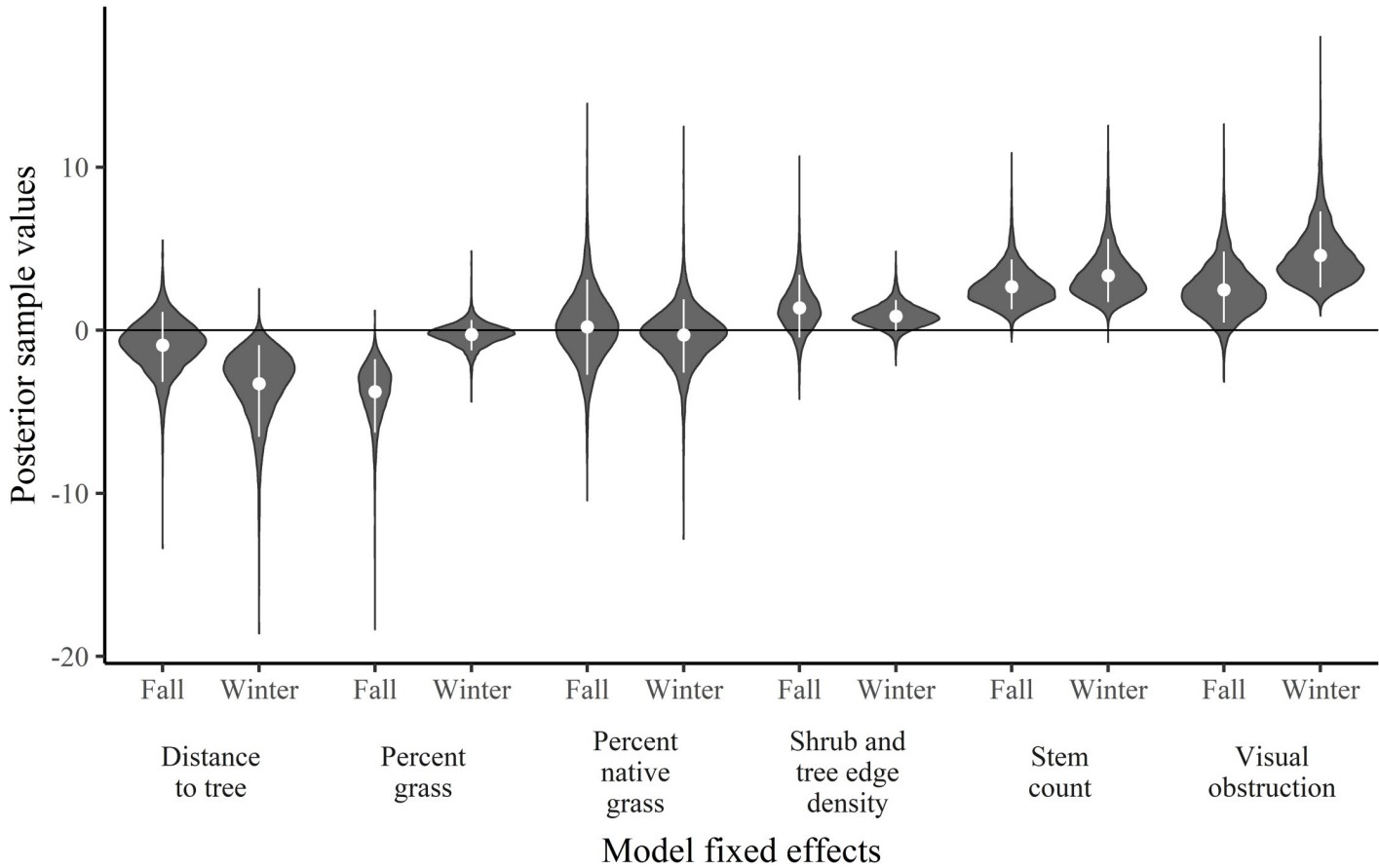

**Fig 2.** Posterior distributions (grey "violins"), means (white dots), and 85% credible intervals (CRI; white bars) from the top model (m1) representing vegetation and management effects on fall (Nov) and winter (Dec–Jan) resource selection of northern bobwhite, Nov 2018–Jan 2019, in southwest Missouri. Model effects are percent grass within the 0.50-m² quadrat frame, percent of native grass within 50 m, quantity of woody stems greater than 1 m in height and within 5 m of the given location, distance (m) from nearest tree edge or individual tree, visual obstruction measured by Robel pole, density of shrub and tree edges within 50 m.

Contrary to our hypothesis, bobwhite selected locations closer to trees, but only in winter. In Ohio, bobwhite selected for woodlots at the home-range order of selection [22]. Similarly, bobwhite selected woodlots over shrubland in New Jersey [27]. Bobwhites may use woodlands when the understory provides early successional woody vegetation [28,45,89,90], likely because the early successional woody vegetation is functionally similar to shrubs [22]. However, trees are known as important habitat for predators of bobwhite such as opossums (*Didelphis virginiana*) and raccoons (*Procyon lotor*) [91,92]. Our research on fall and winter bobwhite survival showed that bobwhite survival increased with distance from trees [93]. Paradoxically, bobwhites selected locations closer to trees, contrary to our hypothesis and the potential fitness benefit of being further from trees. We suggest bobwhites often used shrubs that were in close proximity to trees in areas of more permanent woody vegetation such as fence rows and riparian areas. Alternatively, bobwhites may have selected resources which did not maximize fitness. The strong selection for locations near trees exhibited by bobwhite may indicate trees function as an ecological trap [94], given the negative fitness consequences associated with locations near trees [93].

We found no support for our hypothesis that bobwhite would select locations with greater proportions of native grasses within 50 m in fall, but not winter. Although previous research has shown that bobwhite select for locations in native grass in summer [39,40,95,96], bobwhite in our study did not select for locations with higher percentages of native warm season grasses

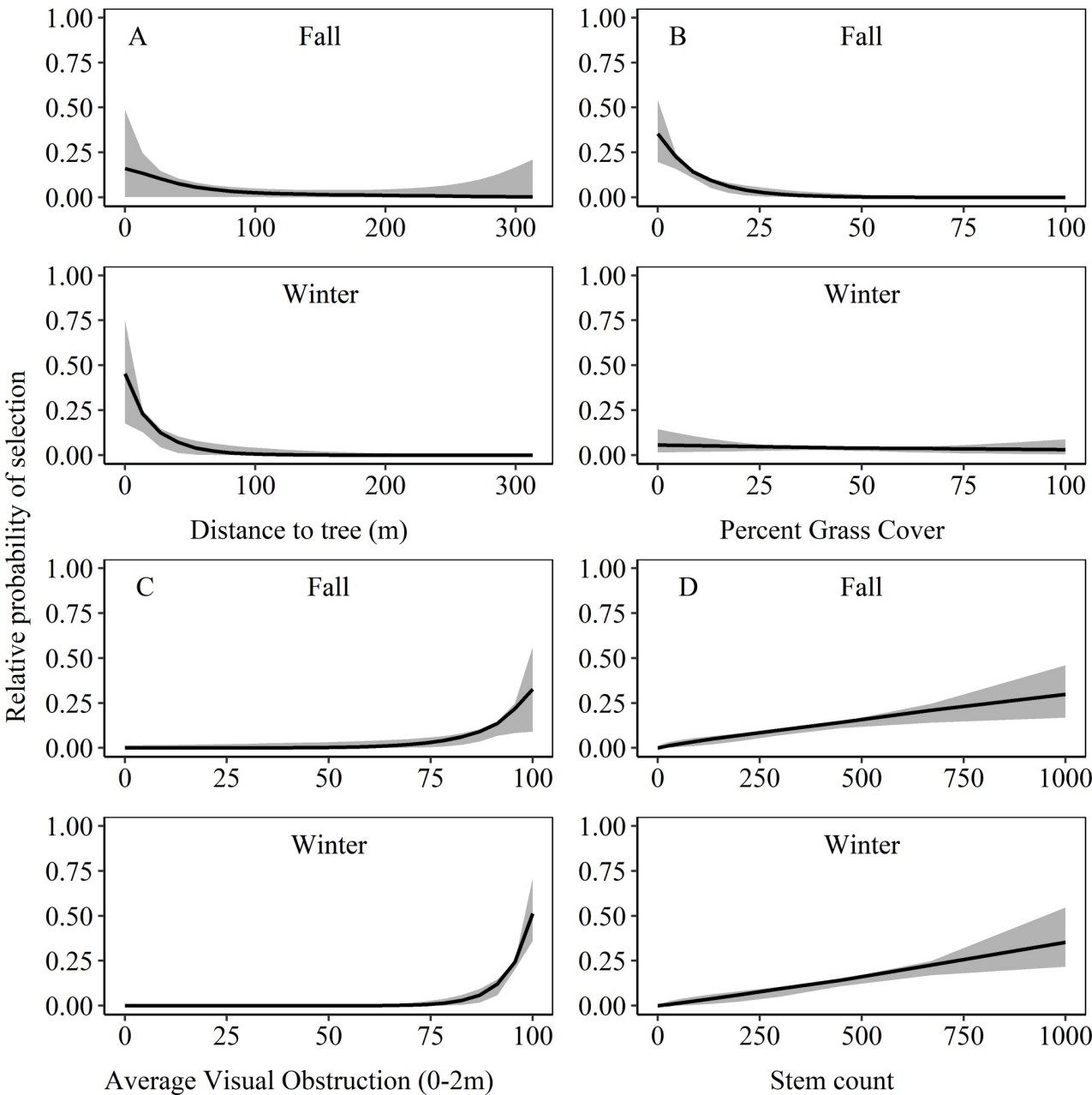

**Fig 3.** Mean relative probability of selection and 85% CRI for Northern bobwhite in fall and winter as function of the A) distance from the nearest tree (m), B) percent grass cover measured within 0.50-m$^2$ quadrat frame, C) average visual obstruction (0–2 m), and D) woody stem count with 5m in southwest Missouri, Nov 2018 –Jan 2019.

within 50 m of their location when compared to all other vegetation types combined. Bobwhite in Ohio selected grasslands when row crops where most abundant, and row crops where grasslands were more abundant [22]. Brooke et al. found that in Kentucky the probability of selection decreased 2% with every 10 m increase from the location to native warm-season grasses [9]. While we do not dispute the importance of native grass, we believe our findings alongside other recent research suggests the importance of native grassland which include patches of shrubs across the landscape for bobwhite.

Previous research has indicated that prescribed burning and grazing manage the structure and composition of vegetation through the removal of litter [46,47] and increased forage abundance and availability [48]. In Kentucky, bobwhites were found closer to native warm season grasses which provided escape and thermal cover throughout the nesting and brood rearing seasons [88]. Brooke et al. found no support for the influence of burning during the previous dormant season on resource selection of non-breeding bobwhite in Kentucky [9]. Fuhlendorf and Engle demonstrated that fire and focal grazing promote heterogeneous grasslands through increased forb abundance, diversity, and structural complexity as well as decreased litter and the decreased prevalence of tallgrasses for 2–3 years post-fire [97]. While prescribed burning and grazing did not substantially influence fall and winter resource selection in our study directly, the disturbance created by prescribed burning and grazing promotes diverse vegetation structure on the landscape. We suggest that conservation practitioners should consider utilizing prescribed burning and grazing at intervals and intensities that promote the creation and maintenance of early successional woody vegetation for bobwhite.

Local structural diversity provided by forbs, grasses, and bare ground is important for bobwhites [37,38,95]. In Texas, average percent forb at used locations during the winter was 16.9% and 18.3% where management regimes were atypical and consisted of short duration grazing and 2-pasture deferred-rotation grazing, respectively, while percent bare ground was 26.3% and 20.0% [98]. We found no support for our hypothesis that bobwhite would select locations with moderate amounts of forb and bare ground. Given the relatively similar distribution of forbs and bare ground across used and available locations, it seems appropriate that there was no selection against forb and bare ground. Bobwhite also selected locations with minimal percentages of grass at the location during the winter, contrary to our hypothesis. Previous research has noted the negative impacts of dense grass on movement and feeding [99–101] and roosting sites [95]. While grasses are a requirement of bobwhite for nesting cover, the value of grass for breeding bobwhite can be replaced by the importance of woody vegetation for non-breeding bobwhite.

Visual obstruction is a measure of vegetation structure provided by all vegetation types that may be indicative of protection and concealment from the weather and predators. Bobwhite in Missouri selected winter roost locations which provided greater visual obstruction (early successional vegetation VOR = 21 cm; native warm season grass VOR = 29 cm) than random sites perhaps due to the increased thermal values of locations composed of taller vegetation [40]. Brooke et al. found coveys selected locations with denser vegetation structure from 1.75 to 2 m above ground [9]. We similarly found that bobwhite selected locations with greater visual obstruction, which supported our hypothesis and previous research.

The amount (i.e., woody edge density) and interspersion (i.e., distance to shrubs) of early successional woody vegetation was the most important contributor to the fall and winter resource selection of bobwhite. While prescribed burning and grazing were not directly supported, these management techniques provided heterogeneity in the form of shrubby thickets dispersed within grassland and set back succession to prevent grasslands from being taken over by shrubs or shrubs succeeding to trees. However, management techniques such as mowing and mechanical brush removal may also be utilized to prevent this succession. Bobwhite populations may be stabilized, or increased, if the amount of usable space available to bobwhite is increased. We suggest that the creation and maintenance of early successional woody vegetation is an important aspect to increasing the amount of usable space for bobwhite throughout the fall and winter.

## Conclusions

Woody vegetation was an important characteristic of locations selected by bobwhite during the fall and winter in Missouri. Specifically, greater quantities of woody stems, indicative of

shrub thickets, and closer proximity to trees increased the relative probability of selection. However, fall and winter survival of bobwhite was lower when bobwhite were located near trees [92], potentially indicating an ecological trap. We suggest that managing disturbance, whether mowing, fire, or grazing, at a return interval and spatial scale that provides high interspersion of shrubs in grasslands while preventing succession to trees, will provide quality fall and winter habitat. Felling mature trees while still maintaining adequate interspersion of shrubs may also prevent an ecological trap. Lastly, while our study focused on fall and winter vegetation, managers should also be cognizant of quail needs throughout their full annual cycle.

## Acknowledgments

N. Burrell, M. Hill, K. Hedges, F. Loncarich, and S. Whitaker provided logistical support. J. Fraser and B. Dijak provided GIS assistance. T. Bonnott, T. Schafer, and E. Schliep provided statistical assistance. E. Sinnott assisted with several aspects of the study. V. Armentrout, M. Conrad, L. Flesher, L. Gilmore, J. Heuschkel, J. Huang, D. King, J. Kuhn, T. Lindsay, L. McElroy, L. Parr, W. Payette, S. Plesh, M. Wheeler, N. Yerden and A. Zak assisted with field data collection.

## Author Contributions

**Conceptualization:** Alisha R. Mosloff, Mitch D. Weegman, Frank R. Thompson, Thomas R. Thompson.

**Data curation:** Alisha R. Mosloff.

**Formal analysis:** Alisha R. Mosloff.

**Funding acquisition:** Mitch D. Weegman, Frank R. Thompson, Thomas R. Thompson.

**Investigation:** Alisha R. Mosloff, Frank R. Thompson, Thomas R. Thompson.

**Methodology:** Alisha R. Mosloff, Mitch D. Weegman, Frank R. Thompson, Thomas R. Thompson.

**Project administration:** Mitch D. Weegman, Frank R. Thompson.

**Resources:** Mitch D. Weegman, Frank R. Thompson, Thomas R. Thompson.

**Supervision:** Mitch D. Weegman, Frank R. Thompson.

**Validation:** Alisha R. Mosloff, Thomas R. Thompson.

**Visualization:** Alisha R. Mosloff.

**Writing – original draft:** Alisha R. Mosloff.

**Writing – review & editing:** Mitch D. Weegman, Frank R. Thompson, Thomas R. Thompson.

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
