## [Decision Letter · Decision Letter 0]

21 Dec 2020

PONE-D-20-33262

Northern bobwhite select for shrubby thickets interspersed in grasslands during fall and winter

PLOS ONE

Dear Dr. MOSLOFF,

Thank you for submitting your manuscript to PLOS ONE. Your manuscript has been assessed by two subject experts and myself, the Academic Editor. All three of us agree that the manuscript is well presented an provides some interesting insight into bobwhite ecology.  However,  the manuscript as submitted does not fully meet PLOS ONE’s publication criteria. Therefore, we invite you to submit a revised version of the manuscript that addresses the points raised during the review process.

Required Revisions

1. Both subject experts expressed concern over lack of clarity in the statistical methods. In addition, Reviewer 1 suggests that the hypotheses being tested might be better addressed with several tweaks to the overall modeling approach. Please address the statistical critique in your revised draft and response.

Recommended Revisions:

1. Please carefully consider all of the points raised by reviewers. Make revisions as deemed appropriate. Please address all comments in the response letter submitted with the revised manuscript.

We look forward to receiving your revised manuscript.

Kind regards,

Christopher M. Somers

Academic Editor

PLOS ONE

Journal Requirements:

2.) In your Methods section, please provide additional location information of the study sites, including geographic coordinates for the data set if available.

3.) In your Methods section, please provide additional information regarding the permits you obtained for the work. Please ensure you have included the full name of the authority that approved the field site access and, if no permits were required, a brief statement explaining why.

4.) We note that you have stated that you will provide repository information for your data at acceptance. Should your manuscript be accepted for publication, we will hold it until you provide the relevant accession numbers or DOIs necessary to access your data. If you wish to make changes to your Data Availability statement, please describe these changes in your cover letter and we will update your Data Availability statement to reflect the information you provide.

5.) We note that Figure 1 in your submission contains map images which may be copyrighted. All PLOS content is published under the Creative Commons Attribution License (CC BY 4.0), which means that the manuscript, images, and Supporting Information files will be freely available online, and any third party is permitted to access, download, copy, distribute, and use these materials in any way, even commercially, with proper attribution. For these reasons, we cannot publish previously copyrighted maps or satellite images created using proprietary data, such as Google software (Google Maps, Street View, and Earth). For more information, see our copyright guidelines: http://journals.plos.org/plosone/s/licenses-and-copyright.

Reviewers' comments:

Reviewer's Responses to Questions

**Comments to the Author**

1. Is the manuscript technically sound, and do the data support the conclusions?

Reviewer #1: Yes

Reviewer #2: Partly

2. Has the statistical analysis been performed appropriately and rigorously? 

Reviewer #1: Yes

Reviewer #2: Yes

3. Have the authors made all data underlying the findings in their manuscript fully available?

Reviewer #1: Yes

Reviewer #2: Yes

4. Is the manuscript presented in an intelligible fashion and written in standard English?

Reviewer #1: Yes

Reviewer #2: Yes

5. Review Comments to the Author

Reviewer #1: This is an excellent study examining resource selection by bobwhite in Missouri. The examination of resource selection by bobwhite in seasons (fall and winter) outside of summer, make this a unique contribution to our understanding of this species. The paper is well written and statistics are robust. I have provided some comments below and I hope you find these useful - thank you for the opportunity to review your paper.

1. It seems as though you had specific hypotheses that bobwhite might be selecting habitat differently in fall and winter (thus including fall and winter as a fixed effect) – this seems like a valid approach and you describe many reasons for this in the introduction. However, by including fall/winter as a binary fixed effect and not modelling resource selection separately, aren’t you constraining your results such that one model with all the same variables is produced for fall and winter (i.e., there is no possibility that a different set of variables could be important in fall and winter – e.g., trees might be important in fall but not at all in winter)? Why not just conduct separate resource selection analyses for data from fall and winter, instead of combing into one analysis?

2. It is unclear how you chose the 6 models displayed in Table 2 – the process of identifying models m1-m6 is not described in the methods. For example it does not appear that you examined a model with all the variables (i.e., a global model) – was this on purpose?

3. It appears as though you went through great lengths to capture and select bobwhite from intensively managed and extensively managed landscapes. But then birds from each of these groups are essentially lumped together for all analyses. Is this because the “discrete choice” modelling takes that information into account and can deal with issues such as different habitat availability etc? I would assume that if bobwhites only used the native habitats in the intensively managed landscapes and the choices they had were all non-native habitats, you would see a “strong” pattern of selection. Whereas if bobwhites used native habitats in the extensively managed landscapes and their only choices were other native habitats you might observe no selection (your used points and random points would have grass at all of them)?And then by combining them you’d end up with a result somewhere in the middle - Is this an issues in the analyses or does the discrete choice modelling tackle that issue?

Minor Comments

Line 65-66: I’m not sure trade-off is exactly the correct term to use here. There may not be a trade-off and habitat selection may be exactly the same in all seasons (or you might be able to manage the landscape such that no habitat is lost at the expense of creating other habitat). Do you mean “to refine management decisions” based on information for other seasons?

Line 71: Is there another term besides “rank” that you could use? That may not be a term that is familiar to everyone. Would “overly tall and dense” be appropriate?

Paragraph starting on line 168: This paragraph is filled with very “jargony” habitat management terms making it difficult to understand (e.g., what is brush hogging, high clipping, what are food plots?). The terms extensively managed and intensively managed also do not seem like particularly common terms – please explain. It appears as though the native grassland landscapes are managed to try to keep them native, whereas the intensively managed landscapes are composed of non-native habitat. There also seems to be a bit of a contradiction in the intensively managed landscapes – you list all non-native habitat types in lines 174-175, but then go on to say that some of these fields were converted back to native prairie, so why not just list native prairie as a cover type?

Paragraph starting on line 240: Some similar issues as my previous comment. For readers not from Missouri these terms are somewhat confusing (what is the difference between native prairie, native grass planting and restored prairie). This will help to understand why you grouped these together.

Line 300-301: I am a bit unclear as to why some of these variables (e.g., agricultural crops, woody cover) were measured via a Daubenmire frame and not just extracted from your landuse/landcover maps? Wouldn’t the landcover mapping be more appropriate?

Line 332: the "NG PG" term should be explained in this paragraph as well. I assume this is an interactive effect?

Paragraph starting on line 517: Could the bobwhite be attracted to trees and shrubs in winter to provide some degree of cover from the elements, but are at higher risk from predators?

Table 1: Given that there is a difference in selection in fall and winter, this table would be more useful if it provided the means and ranges for fall and winter separated rather than lumped together.

Table 2:

- Further explanation is needed in the methods about what the <0.85 and >0.85 posterior distribution indicates. Why is the 0.85 value chosen?

- Make the figure heading clearer by adding the "fall/winter" after the sentence concerning the directions of effect. It’s not clear when you refer to “fall/winter” in the sentence previous that is it referring to how the table is structured.

Reviewer #2: Generally, this manuscript was well-written, easy to follow, and the research was sound. I have a couple major points of criticism. First, they do a good job of detailing the linear model equation, and what each parameter represents. But, how did they decide on the variables within models? They state that they used a VIF threshold for combinations of variables but not how they determine combinations. Was every variable combination checked? What process was used for creating the combinations of parameters? Second, the researchers don’t formally quantify heterogeneity. They measure percent cover of different cover types, and they include disturbance regimes and the land-use history, but they don’t quantify the arrangement of the cover types within their sample areas. There are ways to quantify heterogeneity. Interspersion juxtaposition, perimeter to area ratios, coefficient of variation, etc.. I would avoid using terminology like “heterogeneous” unless it is explicitly defined for the context of your research. As it is, your discussion and interpretation of the significance of heterogeneity is inappropriate and does not correspond to your analyses. You can draw inference from the amount of cover available and used.

6. PLOS authors have the option to publish the peer review history of their article (what does this mean?). If published, this will include your full peer review and any attached files.

Reviewer #1: No

Reviewer #2: No

---

## [Author Response · Author response to Decision Letter 0]

18 May 2021

We have uploaded a response to reviewers file that details our revisions given the academic editor and reviewer feedback.

---

## [Editor Report · Decision Letter 1]

26 May 2021

PONE-D-20-33262R1

Northern bobwhite select for shrubby thickets interspersed in grasslands during fall and winter

PLOS ONE

Dear Dr. MOSLOFF,

Thank you for submitting your  revised manuscript to PLOS ONE. The new version is much improved; I commend the authors for their thorough responses and revisions. There is just one change required that is listed below.

Required revisions:

1. Figure 1 showing the study area is very regional; it should show the larger geographic context in North America. PlosOne is an international journal, and Figure 1 needs to have sufficient map-based information so that readers do not have to look up the study location based on coordinates provided. 

We look forward to receiving your revised manuscript.

Kind regards,

Christopher M. Somers

Academic Editor

PLOS ONE

Journal Requirements:

Additional Editor Comments (if provided):

The authors have done a thorough job of responding to the feedback provided by reviewers; the manuscript reads very well. I commend the authors for their attention to detail.

---

## [Author Response · Author response to Decision Letter 1]

10 Jul 2021

1. Figure 1 showing the study area is very regional; it should show the larger geographic context in North America. PlosOne is an international journal, and Figure 1 needs to have sufficient map-based information so that readers do not have to look up the study location based on coordinates provided. 

We have edited the figure to indicate the study area in relation to the United States.

---

## [Editor Report · Decision Letter 2]

14 Jul 2021

Northern bobwhite select for shrubby thickets interspersed in grasslands during fall and winter

PONE-D-20-33262R2

Dear Dr. MOSLOFF,

We’re pleased to inform you that your manuscript has been judged scientifically suitable for publication and will be formally accepted for publication once it meets all outstanding technical requirements.

Kind regards,

Christopher M. Somers

Academic Editor

PLOS ONE
---

## [Editor Report · Acceptance letter]

9 Aug 2021

PONE-D-20-33262R2 

Northern bobwhite select for shrubby thickets interspersed in grasslands during fall and winter 

Dear Dr. MOSLOFF:

I'm pleased to inform you that your manuscript has been deemed suitable for publication in PLOS ONE. Congratulations! Your manuscript is now with our production department. 

Kind regards, 

on behalf of

Dr. Christopher M. Somers 

Academic Editor

PLOS ONE